# On-Chip Optical Beam Manipulation with an Electrically Tunable Lithium-Niobate-on-Insulator Metasurface

**DOI:** 10.3390/mi13030472

**Published:** 2022-03-19

**Authors:** Linyuan Dou, Lingyun Xie, Zeyong Wei, Zhanshan Wang, Xinbin Cheng

**Affiliations:** 1Institute of Precision Optical Engineering, School of Physics Science and Engineering, Tongji University, Shanghai 200092, China; 1852053@tongji.edu.cn (L.D.); wangzs@tongji.edu.cn (Z.W.); chengxb@tongji.edu.cn (X.C.); 2MOE Key Laboratory of Advanced Micro-Structured Materials, Shanghai 200092, China; 3Shanghai Frontiers Science Research Base of Digital Optics, Tongji University, Shanghai 200092, China

**Keywords:** beam modulator, focus, LNOI, metasurface, on-chip, reflection, tunable

## Abstract

Photonic integrated circuits (PICs) have garnered increasing attention because of their high efficiency in information processing. Recently, lithium niobate on insulator (LNOI) has become a new platform for PICs with excellent properties. Several tunable devices such as on-chip tunable devices that utilize the electric-optic effect of LN have been reported. However, an on-chip electrically tunable beam modulator that can focus or deflect the wave has not yet been developed. In this study, we designed an electrically tunable LNOI metasurface for on-chip optical beam manipulation. With a carefully designed local phase profile, we realized the tunable focusing and reflection functions on the chip. As the bias voltage varies, the focusing length can be shifted up to 19.9 μm (~13λ), whereas the focusing efficiency remains greater than 72%. A continuously tunable deflection can also be achieved efficiently within a range of 0–45°. The beam modulator enhances the ability to manipulate light on LNOI chips, which is expected to promote the development of integrated on-chip photonics.

## 1. Introduction

Photonic integrated circuits are more efficient and safer than electronic integrated circuits with regard to information processing and transmission. Photonic integrated chips might be one of the core technologies in the next generation of information revolution. Numerous materials, such as silicon, silicon nitride, and indium phosphide, can be employed as platforms [1]. However, lithium niobate (LN) is under the spotlight because of its large transparency window, large refractive index, strong second-order nonlinearity, and strong electro-optic effect [2]. LN has become a competitive material in integrated photonics [3].

In recent years, as LN on insulator (LNOI) developed rapidly [4,5], the properties of LN have been gradually investigated. Similar to silicon on insulator, LNOI consists of a substrate made from silicon or LN, on top of which there is a sub-micrometer-thick LN film on a silica buried layer [6]. With the commercialization of LNOI substrates in recent years, large-scale LNOI substrates have provided an excellent platform for integrated photonics, which greatly promotes the research and development of on-chip integrated photonics [2]. LNOI has become a rapidly growing and highly promising integrated photonic platform [6]. Numerous on-chip photonic devices based on LNOI have been reported recently, such as LN low-loss waveguides [7], high Q-factor microring resonators [8,9], Mach–Zehnder interferometer modulators [10], nonlinear optic devices [11,12,13], and metasurface-based devices [14]. Because of the electro-optic effect in LN, which allows the refractive index to vary upon application of an electric voltage, on-chip electrically tunable devices have been widely reported [1,15,16]. Moreover, because LNOI has the structure of a sub-micrometer LN on top of silica, it has a larger refractive index contrast and higher field confinement, making it an excellent electrically tunable platform [2]. Electrically tunable devices can perform functions in several cases. Tunability enhances the control of the light on-chip, which is of great importance in photonic devices [17,18,19,20].

The integrated photonic chip is two-dimensional (2D) and ultracompact. To realize efficient light control in such narrow spaces, researchers are investigating graphene devices [21,22,23,24,25] and metasurfaces. Metasurface is a quickly developing new type of artificial material that is also 2D and can significantly manipulate light while enhancing light-matter interaction [26,27,28,29]. Incorporating a metasurface onto a chip makes it possible to control the on-chip beam at will [30] and realize various complex beam control functions [31]. However, one of the greatest disadvantages of metasurfaces is that their structural geometry, dimensions, and optical properties are fixed. This greatly limits the applications of metasurfaces in the design of tunable devices [32,33]. We have discovered the electro-optic effect of LN that can address this drawback. However, the study of combining the beam-manipulating ability of the metasurface and electro-optic effect of LN still requires further research.

In this study, based on an extensive numerical simulation, we propose an electrically tunable LNOI metasurface for on-chip optical beam manipulation. A series of periodic slits of equal length were used as waveguides. Varying voltages are applied to each unit by two gates such that the refractive index of LN changes separately in each unit. Different refractive indices of each unit result in different phase modulations that add up to a specific phase distribution across the device scale. The phase of each unit varies with voltage, enabling electrically tunable focusing and deflection beam manipulation. The on-chip dynamic focusing function is realized such that the focusing length can be shifted up to 19.9 μm (~13λ) while focusing efficiency remains greater than 72%. The on-chip deflection function was designed without changing the device. A continuously tunable beam deflection was achieved in the range of 0°~45°.

## 2. Materials and Methods

We used a homemade finite-difference time-domain (FDTD) software named Gallop to design and optimize the metasurface on the LNOI chip. The homemade FDTD simulator named Gallop is a three-dimensional electromagnetic field simulation software. The software takes use of the FDTD solutions to calculate photonic problems [34]. We have successfully solved some photonics problems with Gallop [27]. Furthermore, our simulation results were consistent with those obtained by the commercial software from Lumerical. The simulation domain was 3D, rectangular and non-uniformly gridded. The minimum mesh step was set to 50 nm and the simulation time was set to 2000 fs to obtain accurate simulation results. Figure 1a shows a schematic of the designed LNOI on-chip beam modulator. We selected the z-cut LN material (refractive index n_xx_ = n_yy_ = n_o_ = 2.211, n_zz_ = n_e_= 2.138 at λ = 1550 nm [35]), and the refractive index of silica was n = 1.46 at our target wavelength λ = 1550 nm. The transverse electric (TE) wave travels in the positive x direction (polarized in the y-direction). The device as a whole is composed of 19 periodic slits with a periodic width of D = 700 nm, such that the total width of the device is less than 15 μm. The number of the units will increase for more precise wavefront phase control. However, the complexity of the device structure and the difficulty of applying voltages also increase. To find a balance, we have determined that the number of units is 19. The cross-section of a single unit is shown in Figure 1b. Rectangular slits are etched in the middle of the upper LN layer in each unit to form a rectangular waveguide with a width of d = 70 nm to ensure simultaneous efficient transmission and effective electrical control of light. According to numerical simulation, to obtain the best transmission efficiency, the thickness of the upper and lower LN layers of the LNOI platform should be h = 0.5 μm and h_1_ = 2 μm, respectively. The thickness of the silicon dioxide between them was h_2_ = 1 μm. The length of the slit in each unit is L as shown in Figure 1a. On the LNOI chip, the phase will be delayed when the wave travel through the slits. If no voltages are applied, the phase is mainly determined by the length of the slits. Electro-optic effect of LN is used here to modulate the refractive index, so that phase can be modulated by applying voltages to control refractive index variation. 

The device applies voltage V_g_ for each unit by adding ITO material on both sides of the upper LN of the LNOI, as shown in Figure 1b. The lower part of the LN is a layer of ITO material as a whole, and it is covered with ITO material equal to the length of the slits at the top of each unit. Units are independent of each other when applying varying voltages. When the voltages are applied, the refractive index variation of the z-cut birefringent material LN can be expressed as follows [36]:Δn_ii_ = −0.5r_iiz_n_ii_^3^E_z_,(1)
where r_xxz_ = r_yyz_ = 10.12 pm/V, r_zzz_ = 31.45 pm/V denote electro-optic effect coefficients of LN and E_z_ represents the electric field intensity along the z-direction. In other words, the LN refractive index decreases when an electric field is applied in the negative direction along the z-axis. The thickness between the two layers of ITO is the thickness of LN in the upper layer of LNOI, h = 0.5 μm. Therefore, the two layers of ITO can be regarded as parallel plate capacitors. According to the formula for the electro-optic effect, it was observed that every 100 V voltage can change the ordinary refractive index n_o_ by 0.011 and the extraordinary refractive index n_e_ by 0.034.

To achieve on-chip focusing and deflecting beam control, we used a rectangular array of slits along the y direction on the LNOI platform to form a specific phase distribution. To achieve the on-chip focusing function, the phase shift can be defined by the following formula [31]:φ(y) = 2πλ_0_^−1^n_eff_ (F − (F^2^ + y^2^)^1/2^),(2)
where λ_0_ denotes the designed incident wavelength in free space, n_eff_ is the effective refractive index of the LN slit waveguide, and F is the focusing length of the metalens (metasurface with focus functions). Among the previously reported metalenses, some bring different phases through varying sizes and configuration structures [37]; some achieve this by changing the alignment angle of a similar configuration [38]; and others use different lengths of slits to obtain the desired wavefront [31]. In this study, the voltages are applied to each unit of the beam modulator resulting in a high degree of control freedom. We can obtain the desired phase distribution in a simpler manner, more precisely, and more controllably by applying varying voltages to different units. In addition, the phase distribution can be manipulated at high speed by changing the voltage, thus realizing the focusing function of the high-speed tunable on the chip. For the deflection function, the wavefront is relatively simple to obtain by realizing a linear wavefront. Continuous tunable on-chip deflection can be easily realized by changing the phase gradient caused by the voltage.

## 3. Results

The design of the on-chip beam modulator can be divided into three steps. First, the phase and transmittance of the slit waveguide in a single unit are analyzed, and a database is established. Thereafter, design of the device is performed on the basis of the database. Then the device functions of focusing and deflecting are determined.

### 3.1. Units Analysis

To design the beam modulator with superior performance, the parameters of the unit structure were scanned first. The thickness of each part of the LNOI base platform was determined as h = 0.5 μm, h_1_ = 2 μm and h_2_ = 1 μm, respectively, representing the upper LN layer, lower LN layer, and silica layer of LNOI. The periodic width of the unit and the width of the slit waveguide were set as D = 700 nm and d = 70 nm, respectively. Periodic boundary conditions were applied in the y-direction, while perfectly matched layer-absorbing (PML) boundary conditions were applied in the x- and z-directions. Based on the aforementioned parameters, the phase and transmission with the change in slit length without voltage were simulated and the results are shown in Figure 2a. When the slit length reaches L = 20 μm, the phase of the cell structure can cover the range of 0~2π while the transmission is maintained greater than 80% simultaneously. A slit waveguide with a length of L = 20 μm is suitable for the basic structure of the metasurface device. When a different z-axis negative voltage is applied, the ordinary refractive index n_o_ of LN exhibits a decrease in deviation Δn_o_. According to Equation (1) of the electro-optic effect of LN, the voltage required for Δn_o_ = −0.20 is approximately 1828.45 V down the z axis. The maximum Δn_o_ of the previously reported LN tunable filter device is 0.14 [39], which is almost equal to the value determined in this study. Therefore, it is considered that the Δn_o_ value determined in this study is within a reasonable range. The variation of phase and transmission with Δn_o_ was simulated, as shown in Figure 2b. The phase varies almost linearly with a decrease in n_o_, and the transmission is always approximately 80%.

### 3.2. Focusing Function Design

The focusing function design of the LNOI on-chip beam-manipulated metasurface device was performed using the numerical results of unit structure scanning. According to Equation (2), a parabolic wavefront is designed to realize the function of tunable on-chip focusing, that is, a tunable LNOI on-chip metalens is designed. When applied with different combinations of voltages, the metalens can produce varying focusing effects, and the focus can improve significantly. The focusing effect of the metalens at the four different voltage groups is shown in Figure 3. PML boundary conditions were applied in all directions in the simulation. In Figure 3a, Δn_omax_ shown at the top of each figure corresponds to the maximum ordinary refractive index variation due to the voltages applied in all units of the metalens. No voltage was applied to the central unit, and symmetrical voltages were applied to both sides to realize the paraboloid phase distribution, thus realizing the focusing function. Figure 3b shows the distribution of the normalized electric field intensity of y = 0 cross-section versus the x coordinates. The four curves correspond to the metalens in Figure 3a for the four voltage groups. As shown in Figure 3, the focal length of the metalens decreases while the electric field intensity at the focal point increases with the increase in |Δn_omax_| from 0.05 to 0.20. To achieve the beam focusing function, the refractive index of each cell is parabolic distributed, and accordingly, we achieve the function by loading a parabolic distributed bias. No voltage is applied to the unit in the middle of the device. Because of the symmetry of the phase, the voltages applied to the units in the symmetrical position is equal. Therefore, there are 9 units in which different voltages are applied. We also note that it is practically feasible to fabricate electrodes on integrated on-chip photonics [16]. The detailed distribution of refractive index variation among these 9 units is shown in the Appendix A. 

To illustrate the dynamic shift of the focus of the metalens more clearly and characterize the focusing function of the device under different voltage groups, we extracted and analyzed the data, as shown in Figure 4. Four parameters were selected to characterize the performance of the metalens: focusing length, full width at half maximum (FWHM), focusing efficiency (the ratio of light energy to total incident energy in the range of Δy = 3 * FWHM in the focal plane), and transmission ratio at the focal plane. With an increase in the overall voltage, the focusing length and FWHM of the focal spot decrease from 43.32 μm to 23.41 μm and from 1.91 μm to 0.786 μm, respectively. Meanwhile, the focus can be shifted by approximately 19.9 μm (nearly 13λ). Thus, the focusing function improves. In terms of device performance, transmission and focusing efficiency of the metalens under the four groups of voltages are all greater than 78% and 72%, respectively. Thus far, we have designed high-performance tunable metalens on LNOI.

Below Table 1 contains some beam manipulation devices using other materials comparing to the proposed device in this study.

### 3.3. Deflection Function Design

When the structure is not changed, the device can also realize a tunable deflection function on the chip by manipulating the applied voltages, as shown in Figure 5. Because the deflection function is relatively simple, simply applying the linear gradient voltages to the nine units in the middle of the device can bring the linear gradient phase to the incident light on the chip. By changing the voltage gradient, on-chip deflection at different angles can be realized. The incident light in the deflected case is still set to the TE wave propagating in the +x direction. Because the voltage is only applied to the nine units at the center, the light source is set as a Gaussian beam with a half-width of nine units, thus achieving a better and more practical deflection effect. To prove that the deflection function can be manipulated continuously, deflections of 30° and 45° are achieved by setting two specific voltage groups, as shown in Figure 5b,c. In this simulation, PML boundary conditions were also applied in all directions. The maximum variation of the ordinary refractive index Δn_omax_ of LN caused by voltages in the two voltage groups is −0.0475 and −0.095. The detailed distribution of refractive index variation among the 9 units in the middle of the device is shown in the Appendix A. It can also be concluded that the wavefront of the deflection effect is not determined by the phase of the metasurface device. The deflection angle does not change linearly with the voltage gradient because of other unknown effects occurring in our device.

## 4. Discussion

Thus far, we have designed an electrically tunable LNOI metasurface for on-chip optical beam manipulation using the homemade FDTD software named Gallop. By setting periodically equal-length slits on the upper layer LN of LNOI to form a metasurface, the slit waveguides transmit an on-chip optical signal and modulate its phase. By applying different voltages to each unit separately, the electro-optic effect of LN is used to form an electrically tunable local phase profile to realize the focusing and deflecting functions of high-speed electrical tuning on the LNOI chip. In the focusing function, the distance of focus shift is up to 19.9 μm (~13λ) while focusing efficiency remains greater than 72%. The reflection function can be manipulated continuously from 0° to 45°.

Take the electro-optic effect of LN into consideration, all slits are set to the same length as each other. The phase modulation can be manipulated flexibly by changing the voltages applied separately on each unit. Furthermore, because of the flexibility of phase modulation, the device can realize the focusing and deflecting functions without changing its structure. The proposed device provides a novel idea for electrically dynamic tunable beam manipulation based on metasurfaces on an LNOI chip. It enriches and improves the capability of LNOI on-chip beam manipulation and fills the gap in LNOI on-chip tunable beam manipulation devices. It is expected to promote the development of integrated photonics based on the LNOI platform and has broad application prospects.

## Figures and Tables

**Figure 1 micromachines-13-00472-f001:**
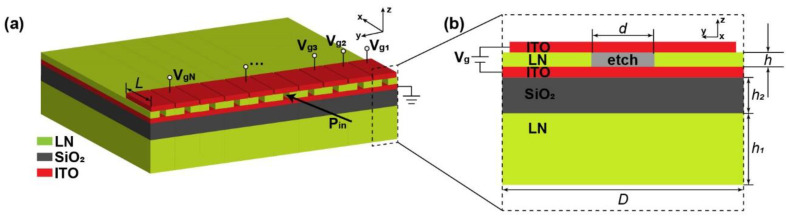
Schematic of the on-chip metasurface device. (**a**) Schematic of the three-dimensional structure of the on-chip metasurface device. The TE wave of wavelength λ = 1550 nm travels in the positive x direction (polarized in the y direction); (**b**) Schematic cross section of the device unit structure. The terms h = 0.5 μm, h_1_= 2 μm and h_2_ = 1 μm denote the thickness of the upper LN layer, lower LN layer, and silicon dioxide layer of LNOI, respectively. The cuboid slits etched in the upper LN layer are in the middle of the unit with a width of d = 70 nm.

**Figure 2 micromachines-13-00472-f002:**
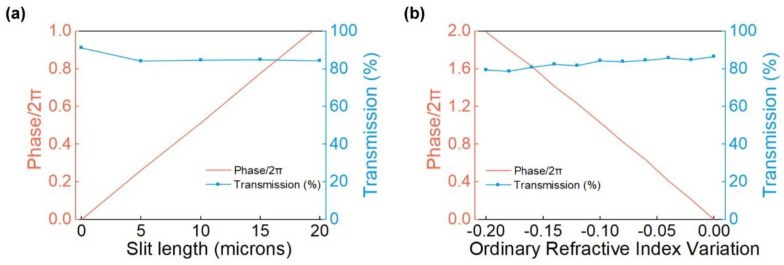
Simulation scanning results of phase and transmission of the element structure. (**a**) Calculation results of the slit waveguide with the length varying from 0 to 20 μm. The phase coverage of the slit waveguide with a length of 20 μm can reach 2π, whereas the transmission is greater than 80%; (**b**) Simulation results of voltage applied under the condition of ordinary refractive index Δn_o_ ranging from 0 to −0.20. |Δn_o_| almost satisfies the linear relationship with the phase, and the transmission is always greater than 80%.

**Figure 3 micromachines-13-00472-f003:**
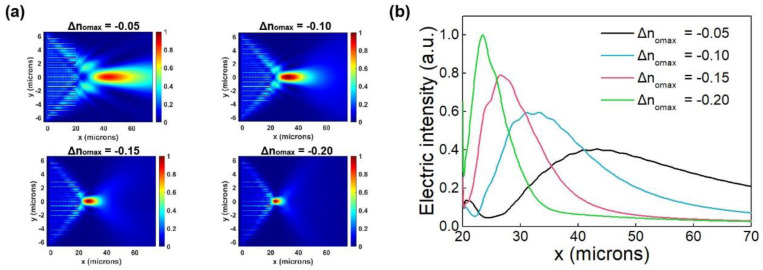
Focusing electric field intensity |E_y_|^2^ distribution. (**a**) Electric field intensity |E_y_|^2^ distribution in the x–y plane of metalens under different voltage groups; Δn_omax_ shown at the top of each figure corresponds to the maximum ordinary refractive index variation due to the voltages applied in all units of the metalens; (**b**) Distribution of normalized electric field intensity of y = 0 cross section versus x coordinates in the metalens. The four curves Δn_omax_ = −0.05, Δn_omax_ = −0.10, Δn_omax_ = −0.15 and Δn_omax_ = −0.20 correspond to the metalens at the four voltage groups shown in (**a**), respectively.

**Figure 4 micromachines-13-00472-f004:**
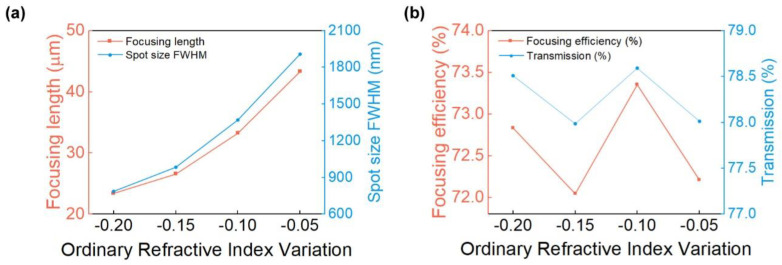
Characterization and comparison of metalens focusing function. (**a**) Comparison of focusing length and focal spot size FWHM of metalens under four voltage groups; (**b**) Comparison of focusing efficiency and transmission ratio of metalens under four voltage groups. The horizontal axis represents Δn_omax_, which is the maximum ordinary refractive index variation caused by voltages in each group. This representation gives a more direct view of the voltage corresponding to different sets of metalens.

**Figure 5 micromachines-13-00472-f005:**
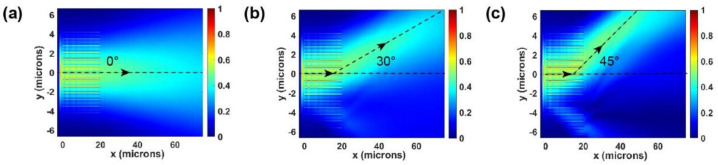
On-chip deflection realized by applying the linear gradient phase with designed voltages. (**a**) |E_y_| with no voltages applied; (**b**) |E_y_| when 30° on-chip deflection is realized. The maximum variation of the ordinary refractive index Δn_omax_ = 0.0475 caused by the applied voltages; (**c**) |E_y_| when 45° on-chip deflection is realized. The maximum variation of the ordinary refractive index Δn_omax_ = 0.095 caused by the applied voltages.

**Table 1 micromachines-13-00472-t001:** Summary of performance metrics for metalenses.

Reference	Wavelength (μm)	Material	F (μm)	FWHM (μm)	Focusing Efficiency	Focusing Length Tunability/μm
[40]	0.43~0.78 (achromatic)	Si_3_N_4_	81.5	2.5~4.5	55% (measurement)	-
[41]	0.4~0.76 (achromatic)	LiNbO_3_	83	1.5~3.2	71% (simulation)	-
[31]	1.55	SOI	25	1.07	79% (simulation)	-
[37]	30	Au, Graphene	161.1~251.5 (tunable)	48.78~60.62	27.15~61.62% (simulation)	90.4 (~3λ)
This work	1.55	LNOI	23.41~43.32 (tunable)	0.786~1.91	72% (simulation)	19.9 (~13λ)

## Data Availability

Not applicable.

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
