# Peer review of "On-Chip Optical Beam Manipulation with an Electrically Tunable Lithium-Niobate-on-Insulator Metasurface"

_micromachines, 2022, doi:10.3390/mi13030472_

Round 1
Reviewer 1 Report
In this paper, the authors propose a design of electrically tunable LNOI metasurface with the function of on-chip optical beam manipulation. The device is composed of periodic LN units with waveguides in the middle, and by adjusting the voltages applied to each unit to change the refractive index, the target phase profile across the device is built up. Comprehensive simulations are given to demonstrate the beam focusing and deflecting capability of the LNOI chip. This work provides a novel strategy for on-chip beam manipulation. However, the following issues should be addressed before it is accepted:
- The investigation is completely based on numerical simulation. In practical situation, the LNOI chip requires additional equipment to apply varying voltages separately to each unit. Thus, will the device become too complicated for the concept of integrated on-chip photonics?
- What dose VΔnomax mean in the legend of Figure 3b? Should it be replaced by Δnomax?
- It is recommended that the authors present the distribution of phase or ordinary refractive index variation among the units required for the focusing and deflecting functions in Figure 3 and 5. The manuscript only gives the the maximum ordinary refractive index variation of the units.
- Does the device design require large numbers of unit structures to improve its phase modulation performance? Since the number of units is limited in the work, does the modulated wavefront accurately conform to the designed wavefront?
- In terms of the potential of LN in integrated photonics and beam modulation, recent advances report the 3D nonlinear photonic crystals fabricated in LN and their applications in nonlinear wavefront shaping and nonlinear holography, which can greatly enrich the functions of on-chip LN devices. Citations of the relevant articles can add to the broad interests of this work. [Quasi-phase-matching-division multiplexing holography in a three-dimensional nonlinear photonic crystal[J]. Light: Science & Applications, 2021, 10:146; Efficient nonlinear beam shaping in three-dimensional lithium niobate nonlinear photonic crystals[J]. Nature Communications, 2019, 10:4193]
Reviewer 2 Report
This paper presents FDTD-simulation based design of electrically tunable LNOI metasurface for on-chip beam manipulation. The proposed device structure may be novel and results may be promising, but it would be difficult for readers to consider this paper is scientifically or technically significant. In addition, I think that the details on device structures and simulation methods are not enough for readers to reproduce author’s idea. 1. Please emphasize the novelty of the proposed device structure 2. Please verify the homemade FDTD simulator. (or describe model calibration. Line 145-146 is not enough for model verification. You may need analytical model to verify your simulation results first.) 3. Please enrich description of the proposed device structure. 4. Please highlight your results compared to other results (e.g. beam manipulation devices using other materials. You may need a table and FoMs for the comparison)Author Response
Please see the attachment.

Round 2
Reviewer 2 Report
The comments were responded well and reflected.